# Aggregation of Italian Lichen Data in ITALIC 7.0

**DOI:** 10.3390/jof9050556

**Published:** 2023-05-11

**Authors:** Stefano Martellos, Matteo Conti, Pier Luigi Nimis

**Affiliations:** Department of Life Sciences, University of Trieste, Via Giorgieri 10, I-34127 Trieste, Italy

**Keywords:** FAIR, open data, identification key, taxon match, Darwin Core

## Abstract

The creation of a coordinated publishing and aggregation system of biodiversity data is a challenging task, which calls for the adoption of open data standards. ITALIC, the information system on Italian lichens, originated from the conversion of the first Italian checklist into a database. While the first version was “frozen”, the present version is continuously updated and provides access to several other data sources and services, such as ecological indicator values, ecological notes and information, traits, images, digital identification keys, etc. The identification keys especially are an ongoing work that will lead to a complete national flora by 2026. Last year, new services were added, one for aligning lists of names with the national checklist, the other for aggregating occurrence data deriving from the digitization of 13 Italian herbaria, forming a total of ca. 88,000 records, which are distributed under a CC BY license and can be exported as CSV files in the Darwin Core format. An aggregator for lichen data will encourage the national community of lichenologists to produce and aggregate further data sets, and it will stimulate data reuse according to the paradigms of open science.

## 1. Introduction

The aggregation and interoperability of biodiversity data are pivotal for facing the challenges of global changes, since together they allow us to use data from individual sources for answering complex global questions [1,2,3]. The main challenge is that of creating a coordinated publishing and integration system, since the isolation of data repositories poses a significant obstacle to the integration and use of biodiversity data [4]. This also calls for the adoption of open data standards [5]. Furthermore, the aggregation and interoperability of biodiversity data have to comply with the 1999 recommendation of the Biodiversity Informatics Subgroup of the Organization for Economic Cooperation and Development’s Megascience Forum [6].

The Global Biodiversity Information Facility (GBIF) [7] is the best-known example of data aggregation and interoperability. The GBIF, since 2001, has been mobilizing primary biodiversity data (i.e., occurrences and data on natural history specimens). Another relevant example, concerning vascular plants, is the World Flora Online [8,9]. Established in 2012 to address the first target of the U.N. Convention on Biological Diversity (the need for an online flora of all known plants), it organizes data on more than 1.4 million plant names. For mammals, the Mammals Diversity Database [10] has, since 2018, provided an updated list of the world’s mammals and is available both online and offline as a freely downloadable package from Zenodo. Another example, which organizes information on marine species, is the World Register of Marine Species (WoRMS) [11], which aims at providing a comprehensive list of names and synonyms of marine organisms, linking them to images, descriptions, documents, and other resources.

As far as lichens are concerned, one of the most relevant efforts is the LIAS data platform [12], which, since 1995, has aimed to organize descriptive and related biodiversity data on lichenized and non-lichenized ascomycetes. The platform has several components, each one focused on a specific goal. One of them, LIAS light, which is specifically focused on lichen traits, achieved its 10,000 species milestone in 2014 [13], and is still growing. Another component, LIAS gtm, integrates GBIF data with lichen traits for achieving the visualization of phenotypic traits via the geographic heatmapping of relative trait frequencies [14]. Another lichen-specific aggregator is the Consortium of North American Lichen Herbaria (CNALH) [15], which aggregates data from herbaria in North and South America, Asia, Europe, and Oceania.

At the national level, one of the most relevant examples is the BLS Database [16], developed and maintained by the British Lichen Society. This database, aiming at organizing records of lichens and fungi in the UK collected in the past 70 years, aggregates ca. 2.3 million records and gives access to species distributions and descriptions. Records are also accessible through the National Biodiversity Network (NBN) Atlas of the UK [17,18] or upon request. 

In Italy, since 2000, ITALIC, the information system on Italian lichens [19], has aimed to publish online the checklist of the lichens hitherto known to occur in the country together with several other types of data, such as ecological indicator values, ecological notes and information, traits, images, etc. 

ITALIC has evolved since its very beginning to become a far more complex aggregator, which is now organizing data from several sources. This paper aims at describing the current structure of, and strategy for, the development of ITALIC, including the steps which are being taken to broaden its role as a central reference for the lichenological community and to strive to achieve full compliance with open and FAIR data principles.

## 2. Materials and Methods

The system of ITALIC is written in PHP 7.0 language, while all the data are hosted in a MySQL database. 

Data are primarily based upon *The Lichens of Italy: A Second Annotated Catalogue* by P.L. Nimis [20], while they are continuously updated on the basis of new findings and taxonomic changes.

Presently, ITALIC operates on the following datasets: (A)Nomenclatural data—three datasets: one for accepted names, one for synonyms (in which a many-to-one relationship with accepted names is allowed), and one for basionyms (which also stores the references to the protologue);(B)Systematic data—a dataset derived from [21,22], which provides the systematic position of each genus in the system;(C)Distribution data—a dataset that reports the presence of each taxon in the administrative subdivisions of the country and the corresponding references;(D)Descriptions—a dataset that is part of the FRIDA [23] system for the production of digital identification keys. It is accessed by ITALIC by means of an API;(E)Ecological data and functional traits—a dataset that contains ecological indicator values, commonness-rarity scores, functional traits, and ecological notes;(F)Herbarium specimens data—13 datasets (at January, 2023) from Italian lichen herbaria. They follow the Darwin Core data scheme [4]. Currently they make up a total of 87,826 records;(G)Digital images archive—a dataset of ca. 52,000 digital images for ca. 6550 taxa. The archive derives from several contributors (metadata are provided with each image). It hosts images for taxa which may occur all over the world, not only in Italy.

The system includes a name-match tool [24] that operates in the query interfaces and/or as an independent instrument, allowing users to align any list of infrageneric taxon names to the nomenclature of ITALIC.

The system is accessible by any web browser, and it is responsive in that it can also provide all its functions on any mobile device.

Some of the authors of this paper are working on a complete e-flora of the lichens of Italy [25], tracking new publications and critically updating taxonomical, nomenclatural, and distributional data, which presently already differ from those published in the checklist of Nimis [20]. All new nomenclatural novelties are critically evaluated before acceptance in ITALIC. The same applies to new records from Italy or from one of its administrative subdivisions: some of them are not accepted if they appear as very improbable, in which case this is stated in the note to the species. For herbaria, a mechanism has been devised by means of which the curators of each herbarium can send to ITALIC a new data table with new records and/or any correction (misidentification) whenever they wish, while nomenclature and synonymizations are managed directly from ITALIC (e.g., whenever a name changes in ITALIC, it also automatically changes in the data tables of the herbaria). Each time a datum is updated, the verbatim datum is stored together with the date of the update and its author. When a taxon name is synonymized, it is stored in the synonyms database, while its place in the system is taken by the new accepted name.

## 3. Results

The first version of ITALIC was discussed by Nimis and Martellos [19]. It originated from the conversion of the text of the first annotated checklist of Italian lichens [26] into a database [27] published online. Since then, the system has evolved strongly, especially in recent years. It now provides two query interfaces (taxon and floristic), access to herbarium data, a national checklist, material for national and regional red lists, a large image archive, and a new tool for matching any taxon name against the adopted nomenclatural backbone (mostly based on [20] but continuously updated). Furthermore, it also provides access to digital identification keys covering practically the whole Italian lichen flora.

### 3.1. Query Interfaces

ITALIC has two main query interfaces. The taxon query interface is quite simple, allowing for the querying of any string of text in a taxon name. As an example, users could submit a query for the string “Xantho” and obtain a list of genera and infrageneric taxa matching the string. The query is key-insensitive, no jolly characters are required, and the string will be matched with any part of each taxon name (accepted names and synonyms) stored in the system. Furthermore, if no match is obtained, the system will make use of a near-match algorithm [24] to look for a match with a similar string and will output a list of text strings that the user can select for performing the query. The output of the taxon query interface is a list of taxa; by clicking on each taxon name, one can access its taxon page.

The same output (a clickable list of taxa) is provided by the floristic query interface, which, however, allows far more complex queries. In the first step, the interface asks whether the user wants to operate solely on taxa actually occurring in Italy or also on those which should be looked for in Italy (since they occur in bordering countries, e.g., across the Alps). In the first case, users also have to select whether they want to query for the whole country and all the ecoregions (see [28] for a detailed description) or for a specific administrative region and/or ecoregion. Finally, users are provided with a query interface where they can select several functional traits and ecological parameters, which can be combined in such a way as to reconstruct different ecological scenarios, thereby obtaining “virtual relevés”, which may even simulate presence–absence data in a single vegetation plot. The predictivity of “virtual relevés” was tested by comparison with real data [29] and proved to be high. The output of a query is a page that at its top reviews all the parameters inputted by the users, followed by statistics on the infrageneric taxa matching the query. Figure 1 shows the results of a query for the lichens potentially occurring on the facades of the Greek temples in Agrigento (Sicily, South Italy). The parameters were Sicily (administrative region), Mediterranean belt, saxicolous, calcareous (basic) substrata, high irradiation, and very limited water availability. The result is a list of 45 taxa, for which a series of statistics are reported. Each taxon name links to the corresponding taxon page. Once the Italian lichen e-flora is completed (see later), instead of a list of species, the users will have at their disposal an identification key for all species potentially occurring in a “virtual habitat”.

### 3.2. Taxon Pages

Taxon pages aggregate all the available data about a taxon (Figure 2). They start with the accepted taxon name and the basionym, if any, with reference to the protologue. The following sections list synonyms, distribution in the 20 administrative subdivisions of the country with literature references for all records after 1993 (when the first annotated checklist was printed), a description, and a note.

In the following part, some functional traits and ecological indicators values are reported, together with the commonness–rarity status of each taxon in each ecoregion of Italy [28]. Furthermore, a push button leads to a “Classification” page (Figure 3), in which, for each genus, a phylogenetic tree [22] and a link to a digital identification key (see later for details) are provided. Another push button leads to a reference query interface.

Another block exposes a predictive distributional map, built by combining presence in the administrative subdivisions of the country with commonness/rarity in the ecoregions (for details, see [28]), and a web-GIS viewer for the distribution of herbarium samples aggregated into the system. Herbaria can be included/excluded from the visualization by clicking on the corresponding checkboxes.

The last part of the taxon page shows all the images stored in the archive, each one with its metadata and license.

### 3.3. IDkeys

Since the late 1990s, within the framework of the Dryades project [30], the research group of the Department of Life Sciences, University of Trieste, in cooperation with other research groups from Italy and abroad, has developed several digital identification keys by using the FRIDA package (FRiendly IDentificAtion) [23]. In 2018, a new effort to build an interactive lichen flora of Italy was initiated [25]. Currently (March 2023), 126 digital dichotomous nationwide keys including ca. 450 genera have been produced, which are accessible in the “Identification keys” section of ITALIC or from the taxon pages. Once all genera are included, they will be integrated into a general key to all the lichens of Italy, which will be published both online and in paper form. Some large, complete keys are already available for the lichens of the Trieste Karst region in Italy and Slovenia (604 species), the lichens of the Alpine belt of the Italian Alps (1320 species), and the lichens of northern Italy (2753 species). These latter keys are also provided with a multi-entry query interface which greatly speeds up the identification process, since a shorter dichotomous key can be obtained for the reduced set of species that share a given set of ecological, morphological, or chemical characters.

### 3.4. Aggregation of Natural History Specimen Data

One of the most relevant novelties of ITALIC seen since 2022 is the aggregation of data coming from the digitization of natural history specimens. Until now, ITALIC presented only data and images produced by the research group of Trieste, which is in charge of updating and maintaining the whole system, although several digital images were provided by authors from other institutions in Italy and abroad. In 2022, a new project to involve all the research groups in Italy that were digitizing their natural history collections was started. The idea was to present in the taxon pages of the system herbarium data in the form of web-GIS maps and allow their download by users. The process started with the Italian samples of the TSB herbarium, which hosts a total of ca. 40,000 specimens, 25,796 of which were collected in Italy. To prepare the dataset, and to align it to the Darwin Core (DwC) standard scheme, several steps were carried out: (A)Review of the original database to check for incongruences (e.g., impossible dates, swished fields, etc.);(B)Review of nomenclature. Original names were aligned with the current nomenclature by means of the name-match tool included in ITALIC. Original names were preserved (using the DwC concept “verbatimIdentification”), while accepted names were added to the dataset using the DwC concept “scientificName”;(C)Georeferencing. No records were originally georeferenced. Thus, appropriate effort was devoted to transforming localities into coordinates using the point–radius method, following the best practices of Chapman and Wieczorek [31]. Uncertainty (the radius) is expressed in meters. In this process, extreme care was taken in order to not underestimate uncertainty. In cases of doubt, a wider uncertainty was adopted;(D)Dates. From original dates (stored as DwC “eventDate” concept), the year alone was extracted and stored in the “year” concept;(E)Harmonization of substrate metadata. Since substrata, when specified on the envelopes, were expressed in a wealth of different formats and languages, they were harmonized in order to be all written in English and in a more or less standardized format;(F)Altitude. Since altitude was expressed in a single concept, but alternatively as a single number or as a range, the concept was split into the two DwC concepts “minimumElevationMeters” and “maximumElevationMeters”.

To facilitate and automatize the preparation of the dataset, a tool was developed following the concept of the GBIF Integrated Publishing Toolkit (IPT) [32]. After mapping the dataset against the Darwin Core concepts, the new tool performs a series of checks to test whether the data are fully compliant with the standard. These checks include looking for missing mandatory data or invalid values. Furthermore, the tool checks whether values are written in the correct format. If discrepancies or errors are found, the tool provides feedback highlighting which issues need to be addressed.

After the upload of the dataset, each record was assigned a CETAF id [33] as a global unique identifier (GUID). CETAF GUIDs are identifiers which are both human- and machine-readable. When users try to access a record by typing its CETAF GUID into a web-browser, they are redirected to a web page, whereas when a software system tries to use the same GUID, it obtains an RDF-encoded metadata object [34]. 

The entire dataset was made available both in the herbaria section and in the taxon pages of ITALIC. In each taxon page, all the occurrences are extracted and plotted on a web-GIS map created by means of Leaflet [35]. The herbaria section allows for more complex queries on the whole dataset (e.g., all species collected from *Fagus sylvatica*). The results are returned in dynamic web pages and can be downloaded as a CSV file in DwC format.

The dataset has been published in the form of a data paper [36], so that it can be properly cited.

The digitization of the TSB herbarium was the starting point for the aggregation project. Several other Italian lichen herbaria, institutional or private, in which a digitization effort was being carried out, were contacted with a proposal to aggregate their specimen data in the system as separate datasets. For each of them, the process described above was replicated (except for the georeferencing), after which a reviewed dataset was provided to every single herbarium for further review and approval. As a further step, those in charge of the dataset were then asked to georeference the occurrences, providing them with proper guidelines. At the end of the process, the georeferenced datasets were aggregated, and the author(s) were requested to publish a data paper to provide a proper means of citation. The publication of data papers is ongoing and will be probably concluded by the end of 2023. A total of 87,826 specimen records from 13 modern herbaria were aggregated: CLU, FI, GDOR, GE, HLUC, ORO, SI, TO, TSB, and the private herbaria of G. Gheza, J. Nascimbene, S. Ravera, and W. von Brackel (Table 1).

The number of specimens and taxa per herbarium is shown in Figure 4. 

The geographical distribution and the temporal distribution of the aggregated records is shown in Figure 5.

The numbers of specimens for each phylum, class, order, family, and genus are graphically visualized in a Krona graph [37] (Appendix A).

## 4. Discussion

Since its original release in 2000, ITALIC has been an important source of information not only for the Italian lichenological community but also for lichenologists abroad. Its pages are accessed every day by an average of ca. 100 unique visitors, with ca. 600 page loads. Furthermore, the portal has a percentage of ca. 80% of returning visitors, thus highlighting that it is seen as a reliable and useful resource by the vast majority of its users. Users come from all over Europe (more than 50% of users come from outside Italy), and several of them are from other continents. New observations and novel findings at regional and national level have been constantly added to the portal each time they have been published on scientific journals, as in the case of the *Notulae to the Italian flora of algae, bryophytes, fungi and lichens*, which are published every six months in the journal *Italian Botanist*. This continuous update, which is carried out by the research group of the Dryades project, also involves the systematics, nomenclature, and ecology of each taxon. Furthermore, the whole system is constantly under development with the addition of novel features and general improvements.

The idea of making ITALIC an aggregator for occurrence data deriving from the digitization of natural history collections arose at the beginning of 2022. The TSB lichen collection was thus used as a test bed for understanding the level of challenge of such an idea, and the georeferencing process was highlighted as the most time-consuming activity. Thus, the point–radius method was chosen due to its simplicity and its efficiency in expressing the uncertainty of the point. Given the interest of the holders of other relevant collections, other datasets were aggregated in the new version of ITALIC (7.0), which was made available online by the end of September 2022.

All the occurrence data aggregated in the system are distributed under a Creative Commons license (CC BY) and can be used by anybody, with the only constraint being the provision of a proper citation. However, as far as citations are concerned, the approach adopted for ITALIC is different from that adopted, for example, by the GBIF. In the latter case, users are provided with a DOI (digital object identifier) for any dataset they download. The DOI always allows users to return the metadata of the dataset and also track the owners of the original datasets from which the data originated. This approach is necessary for a system which aggregates tens of thousands of datasets. However, in the case of ITALIC, the number of datasets which are and will be aggregated is far more limited. Even if all the lichen collections in the country contributed data to the system, only a few dozen datasets would be collected. Thus, a query to the system would normally return a dataset containing data from few original datasets, which can then easily be cited in a paper. For this reason, each dataset aggregated in ITALIC will be published as a data paper in a journal, and its citation will be provided each time a user retrieves its data (whether all or only part of them). This approach will be implemented as soon as all data papers are published, probably by the second half of 2023.

The other most noteworthy resources which have been added to the system in the last two years are the digital identification keys and the names alignment tool. The identification keys are an ongoing project which will see a first conclusion in 2026, thereafter being constantly improved by the addition of new taxa or the updating of the nomenclature.

The names alignment tool was created by reusing the one released on FlorItaly, the portal to the flora of Italy [38], and developed in 2021 [24]. The algorithm was slightly changed, but its core functions remained the same. It can be used to align any list of lichen names to the currently accepted names. Alignment is automatic when a perfect match is achieved; otherwise, users are always asked to choose among possible matches listed in order of decreasing matching scores.

ITALIC differs from other experiences in data aggregation and interoperability for several reasons. 

The GBIF (to which ITALIC contributes with herbarium data) focuses on occurrence data (i.e., observations and specimen data). The World Flora Online [8] and the Catalogue of Life [39], on the contrary, are focused on nomenclatural data; in both cases, few other data are provided on taxa, other than names and synonyms. More similar to ITALIC are the BLS database [16], the CNALH website [15], and LIAS [12]. The BLS database, which includes descriptions, maps, images, and keys, focuses on species occurring in the British Isles and publishes keys in a static form as PDF files, while the keys of ITALIC are available online and are updated constantly. The CNALH website focuses on herbarium specimens in North America, providing descriptions and a searchable image archive with no keys. The LIAS is a global initiative, which organizes functional traits, morphological and chemical characters, and nomenclatural data, providing a tool for the rapid identification of lichens with a free-access query interface. ITALIC differs in scope (having a narrower geographical coverage) but especially in the fact that all the query interfaces, including the online keys (both dichotomous and multi-entry), were designed following the best practices developed in the framework of the European Project KeyToNature [40], coordinated by the University of Trieste, for optimizing their usability and user-friendliness.

## 5. Conclusions

The existence of an aggregator for lichen data in Italy is a relevant resource for the national community of lichenologists, since on one hand, it stimulates further digitization of natural history collections, and on the other hand, it stimulates the reuse of data for novel, collaborative research. 

The general positive feedback received since the release of this new version of the system has stimulated new efforts in the same direction. The first one will be the widening of the aggregation to observation data (e.g., those deriving from phytosociological reléves of lichen vegetation). These data will be flagged in a distinctive way so that they will be easily distinguishable from specimen data (which, contrary to most observation data, are falsifiable).

Another development will be that of at least making occurrence data and digital images FAIR (findable, accessible, interoperable, and retrievable). At the moment, occurrence data have been tested against a FAIRness assessing tool [41], and results are at an initial stage. Coping fully with the FAIR principles is a challenging task but also the most relevant improvement that will be carried out in the near future.

## Figures and Tables

**Figure 1 jof-09-00556-f001:**
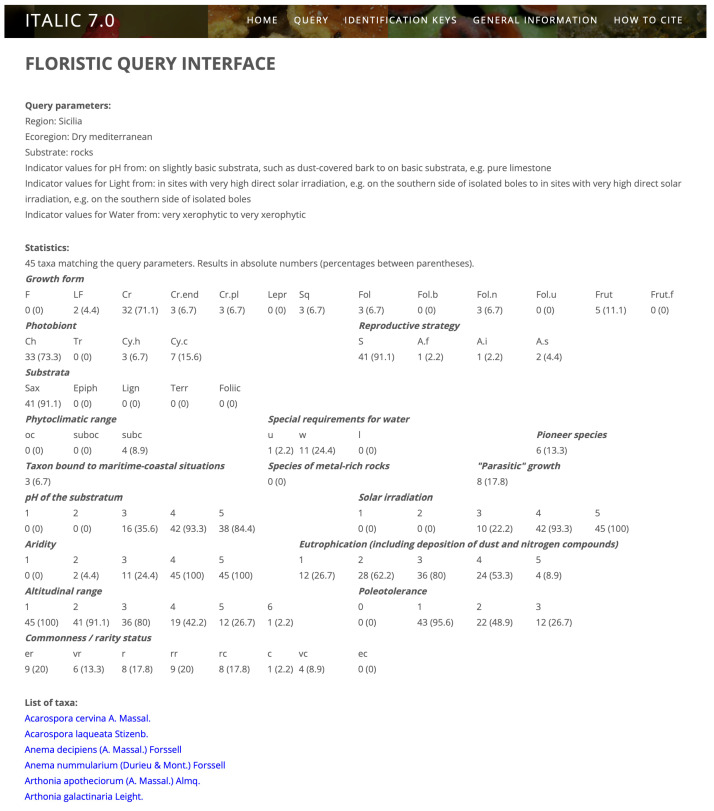
Result of a floristic query.

**Figure 2 jof-09-00556-f002:**
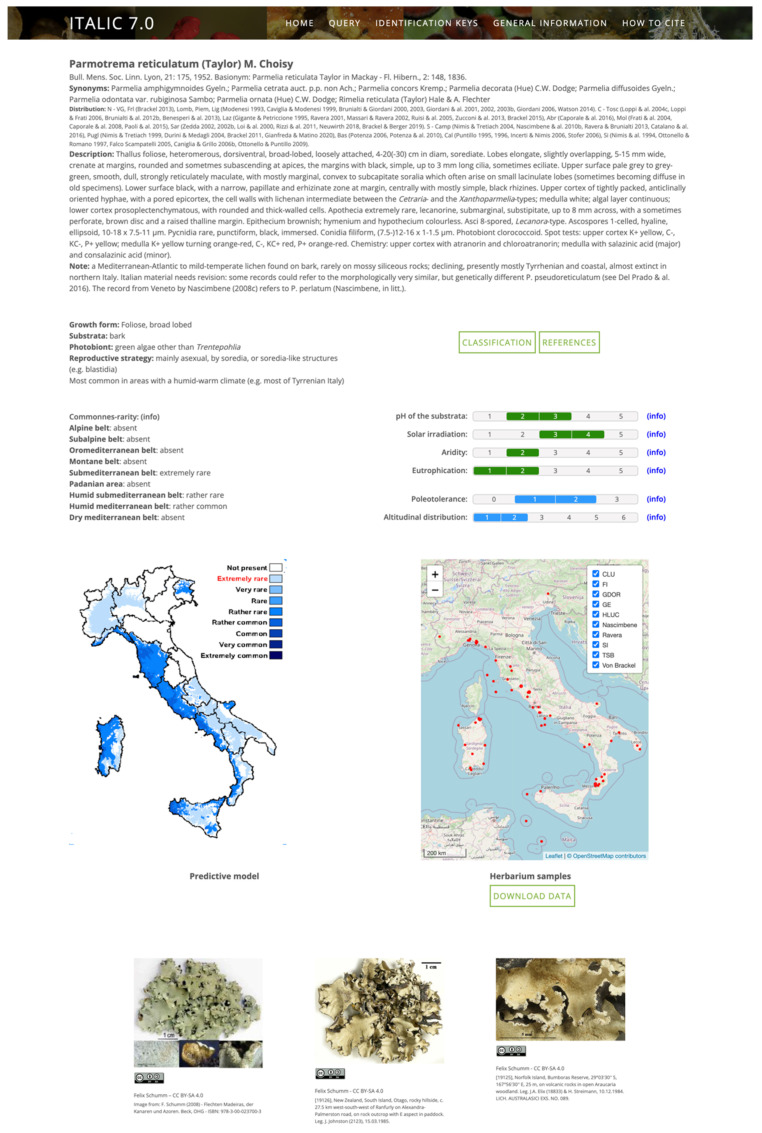
Taxon page of *Parmotrema reticulatum* (Taylor) M. Choisy.

**Figure 3 jof-09-00556-f003:**
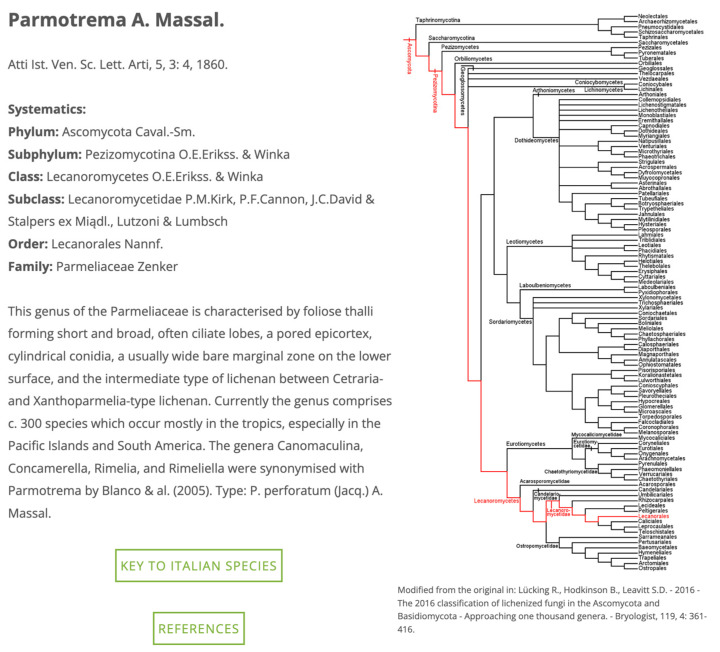
Taxon page with the phylogenetic tree for the genus *Parmotrema* A. Massal [22].

**Figure 4 jof-09-00556-f004:**
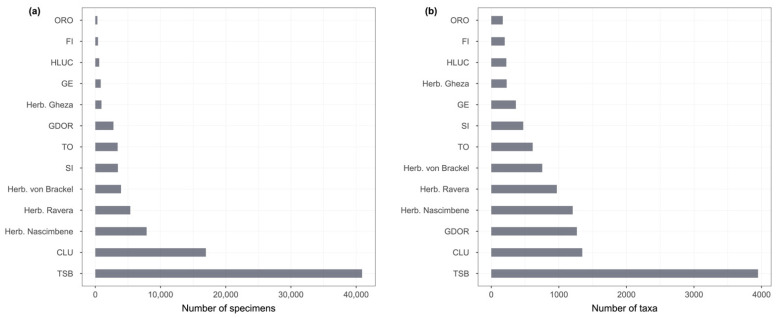
(**a**) Number of specimens in each herbarium; (**b**) Number of taxa in each herbarium.

**Figure 5 jof-09-00556-f005:**
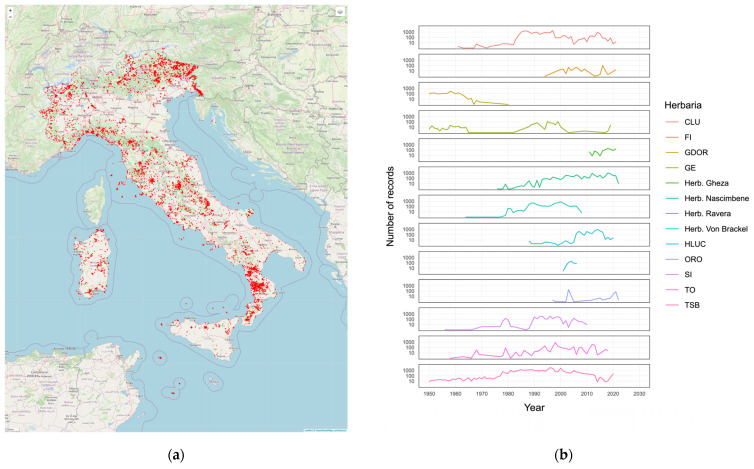
(**a**) Geographic distribution of georeferenced records; (**b**) temporal distribution of records for each herbarium.

**Table 1 jof-09-00556-t001:** Herbaria aggregated in ITALIC, with Index Herbariorum code (if present) and number of records.

Herbarium	Index Herbariorum Code	Number of Records
Erbario del Museo Civico di Storia Naturale Giacomo Doria	GDOR	2782
Erbario Lichenologico Fiorentino	FI	416
Erbario Lichenologico Università della Calabria	CLU	16,956
Flora Montis Oropae	ORO	320
Herbarium Gheza		948
Herbarium Lucanum	HLUC	600
Herbarium Nascimbene		7871
Herbarium Ravera		5363
Herbarium Universitatis Genuensis	GE	831
Herbarium Universitatis Senensis	SI	3460
Herbarium Universitatis Taurinensis	TO	3428
Herbarium Universitatis Tergestinae	TSB	40,908
Herbarium von Brackel		3943

## Data Availability

Not applicable.

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
