# Peer review of "Aggregation of Italian Lichen Data in ITALIC 7.0"

_jof, 2023, doi:10.3390/jof9050556_

Round 1

Reviewer 1 Report

Although this paper is merely descriptive and not experimental, it describes the current structure and the strategy of development of the online lichen database ITALIC.  This paper and ITALIC will be very useful for  the lichenological community. 

The paper is very well written, I do not have comments.

Author Response

Thank you very much for your appreciation.

Reviewer 2 Report

Dear Authors, 

congratulations! you made a great work!

The paper is an original research study on the Aggregation of Italian lichen data in ITALIC 7.0.

The Authors made a great work in terms of methodology and the paper sounds scientific and well written.

However, some improvements are mandatory before acceptance.

The abstract is well written, complete and summary in its various aspects. The keywords are complete and appropriate.

I believe that the introduction offers a lot of information on what is covered in the manuscript, on the background and on the reason that prompted the authors to proceed with this research.

The materials and methods section is concise and complete. The main indications on the research methodology and on the tools used are specified.

The results are well written, comprehensive and offer much information about the study in question. the images are indicative of the software used and allow a better understanding of the terminology used in the various sections. The tables are precise and concise.

Discussion: this section is complete and evaluates the outcome of different papers present in literature. The overall is comprehensive, concise and complete in its various aspects.

Conclusions are concise and clear.

Bibliography should be formatted respecting the journal’s requirements and no improper citations are evidenced.

Figures and labels are clear and easy to comprehend.

English is clear and easy to understand.

I think it is of good quality, written in a simple and clear way.

Author Response

Thank you very much for the appreciation of our work. We tried to improve the new version by carefully English-editing the text, cross-checking again the References (and correcting a few typos), by changing 2 figures (Figs. 2 and 3 showing the same content but selecting a species whose distributional data showed a more distinct pattern), and adding two paragraphs: 1) at the end of the Materials and Methods section, devoted to procedures followed for updating the data, and, 2) at the end of the discussion, devoted to the main differences between ITALIC and other similar websites.

Reviewer 3 Report

Stefano Martellos et al.'s manuscript entitled: Aggregation of Italian lichen data in ITALIC 7.0 presents a valuable study regarding the ITALIC data. The research is well-structured, well-conducted and well-written, but the discussion section must be extended.

Please, extend the discussions by making the comparison of the presented data with the other aggregation and interoperability of biodiversity data already mentioned in the Introduction section, as the subject offers a lot of information.

Generally, the English language is fine, but some transition words are not suitably used

Author Response

At the end of the discussion, we added a new section devoted to the main differences between ITALIC and other similar websites.

This is the newly added section:

ITALIC differs from other experiences in data aggregation and interoperability for several reasons. The GBIF (to which ITALIC contributes with herbarium data) focuses on occurrence data (i.e. observations and specimen data). The World Flora Online [8] and the Catalogue of Life [39], on the contrary, are focused on nomenclatural data; in both cases, few other data are provided on taxa other than names and synonyms. More similar to ITALIC are the BLS database [16], the CNALH website [15] and LIAS [12]. The BLS database, which includes descriptions, maps, images and keys, focuses on species occurring in the British Isles and publishes keys in a static form as pdf files, while the keys of ITALIC are available online and are updated constantly. The CNALH website focuses on herbarium specimens in North America, providing descriptions and a searchable image archive, with no keys. The LIAS is a global initiative, which organizes functional traits, morphological and chemical characters and nomenclatural data, providing a tool for the rapid identification of lichens with a free-access query interface. ITALIC differs in scope (having a narrower geographical coverage) but especially in the fact that all the query interfaces, including the online keys (both dichotomous and multi-entry) were designed following the best practices developed in the framework of the European Project KeyToNature [40], coordinated by the University of Trieste, for optimizing their usability and user-friendliness.

Reviewer 4 Report

The Authors provide a very interesting instrument to scientific community. The amount of information available is really enormous and continuously up-to-date. In my opinion it is an excellent way to exploit the extraordinary potential of data resources, improving the previous version of ITALIC.

Just some really minor remarks:

Line 31: eliminate comma before “is the best…”

Line 81 “stores”

Line 89: please correct “one ecological notes”

Line 189 “were provided” instead of “were provided t”

See the minor remarks suggested.

Author Response

Thank you very much for the appreciation of our work. All of your suggestions were incorporated into the new version.

Reviewer 5 Report

Lichens are relatively poorly studied fungal groups. It is quite nice to have such an aggregated database that integrate data from other specialized databases after their authorizations.

However, I'm not familiar with "The Lichens of Italy", therefore, there are some of my concerns about the ITALIC:

1. How does the database update the newly generated information of Lichens? Are there any people keep tracking new publications regularly?

2. Is the nomenclatural data in ITALIC keep updating with international fungal nomenclatural database, such as IndexFungorum and MycoBank?

3. Is the systematic data ITALIC uses keep updating the newly generated sequence from GenBank and their phylogenetic placement from recent publications?

4. Is the information of herbarial specimens keep updating in case of misidentification or synonymization?

5. Does ITALIC verify the data itself, or just record any relevant data in the public?

6. the "t" should be deleted at the end of line 189.

Author Response

You raised a series of important questions concerning the continuous updating of the system, which we had overlooked in the first version of the paper. All of your points have been now dealt with in a section at the end of the Materials and Methods section.

1 & 2: Prof P.L. Nimis, who is preparing an e-flora of the lichens of Italy (Nimis & Martellos 2020), regularly keeps tracking new publications, critically updating the nomenclature; sometimes the updates occur in ITALIC before they appear in IndexFungorum.

3: The systematic data of ITALIC keeps updating the phylogenetic placement of lichens from recent publications (the systematic scheme which is online now is already different from that published in the checklist of Nimis 2016).

4: For herbaria, we have devised a mechanism by means of which the curators of the single herbaria can send us a new data table with new records and/or any correction (misidentification) whenever they wish, while nomenclature and synonymizations are managed directly from ITALIC. E.g. whenever a name changes in ITALIC, the it automatically changes also in the data tables of the herbaria (while the verbatim name is always preserved).

5: All new nomenclatural data are critically evaluated before insertion in ITALIC, e.g. some nomenclatural novelties are not accepted because they appear not sufficiently justified; the same applies to new records from Italy or from the different administrative subdivisions of Italy, e.g. some literature records are not accepted because they appear as very improbable, in which case this is stated in the note to the species. 

6: Fixed

Round 2

Reviewer 3 Report

Thank you for your reply.

The manuscript needs only minor revision regarding the English language.